# Circulating Folate Concentrations and Risk of Peripheral Neuropathy and Mortality: A Retrospective Cohort Study in the U.K

**DOI:** 10.3390/nu11102443

**Published:** 2019-10-14

**Authors:** Tom Taverner, Francesca L. Crowe, G. Neil Thomas, Krishna Gokhale, Rasiah Thayakaran, Krishnarajah Nirantharakumar, Yusuf A. Rajabally

**Affiliations:** 1Institute of Applied Health Research, College of Medical and Dental Sciences, University of Birmingham, Prichatts Road, Birmingham B15 2TT, UK; T.Taverner@bham.ac.uk (T.T.); G.N.Thomas@bham.ac.uk (G.N.T.); K.M.Gokhale@bham.ac.uk (K.G.); R.Thayakaran@bham.ac.uk (R.T.); 2Aston Brain Centre, Aston University, Aston Triangle, Birmingham B4 7ET, UK; y.rajabally@aston.ac.uk

**Keywords:** electronic health records, epidemiology, folate, nutrition, peripheral neuropathy

## Abstract

Background: Folate deficiency may increase the risk of peripheral neuropathy but there is a paucity of data from large prospective studies examining this association. Methods: Longitudinal analysis of electronic health records in The Health Improvement Network (THIN), a U.K. primary care database including 594,338 patients aged 18–70 years with a folate measurement and without a history of peripheral neuropathy. Results: After a mean follow-up of 3.71 (standard deviation (SD) = 3.14) years, 1949 patients were diagnosed with peripheral neuropathy and 20,679 patients died. In those <40 years, compared to patients with folate ≥13.6 nmol/L, those with folate <6.8 (deficient) and 6.8–13.5 nmol/L (insufficient) had a hazard ratio (HR) for peripheral neuropathy of 1.83 (95% confidence intervals (CI) = 1.16–2.91) and 1.48 (95% CI = 1.04–2.08), respectively. There was no significant association between folate and peripheral neuropathy among those aged 41–70 years. Compared to patients with folate ≥ 13.6 nmol/L, folate <6.8 nmol/L was associated with a greater risk of death among all ages. Conclusion: Folate deficiency and insufficiency was associated with a greater risk of peripheral neuropathy among younger patients. This investigation should be replicated in other large datasets and it may be important to monitor peripheral neuropathy incidence after the introduction of mandatory folic acid fortification of flour in the U.K.

## 1. Introduction

Peripheral neuropathy is a term given to indicate a disorder of the peripheral nervous system. It can involve sensory and/or motor fibres, and is relatively common, with an estimated prevalence of 8% in adults older than 55 years [1]. The most common type of peripheral neuropathy is polyneuropathy, with the most common form of polyneuropathy involving the length-dependent axonal part of nerve cells where the process initially involves the axon of the distal portion of the longest nerves with subsequent gradual proximal extension via a “dying back” process. In such cases, the predominant feature is numbness in the lower and upper extremities, paraesthesia and pains, and there is usually a delayed involvement of motor fibres, which causes weakness. As result, polyneuropathy can cause considerable discomfort, pain, functional disability and reduction in the quality of life. The most common causes of polyneuropathy are diabetes, excessive alcohol consumption and deficiencies in certain vitamins, but about 25% are considered idiopathic [2].

Folate is a water-soluble B vitamin, and in the U.K., good dietary sources are green vegetables, such as broccoli, citrus fruit and foods voluntarily fortified with folic acid, including some breakfast cereals [3]. Deficiency in folate usually results from an insufficient intake but may also be caused by impaired absorption, increased requirements, use of certain medications and chronic alcoholism [4]. Folate deficiency causes neurological manifestations including spina bifida and anencephaly during foetal development [5], and has also been linked with myelopathy [6]. Patients with folate deficiency can develop peripheral neuropathy [7], and folate levels are commonly tested for in the clinical context, although there is little evidence from prospective cohort studies to support this hypothesis. Given the recent decision of the U.K. government to introduce a policy to fortify flour with folic acid [8], it is important to determine whether isolated folate deficiency may have a role in polyneuropathy before this occurs, as folate deficiency is projected to decrease.

In this study we aimed to assess the relationship between folate deficiency and the risk of peripheral neuropathy of an otherwise undetermined aetiology in a large U.K. population younger than 70 years. A secondary objective was to assess the association between serum folate and all-cause mortality.

## 2. Materials and Methods

### 2.1. Study Design and Setting

This was a retrospective cohort found in The Health Improvement Network (THIN) primary care database, which contains health records for more than 11 million patients from over 600 general practices (GPs) in the U.K. The distribution of age, sex, prevalence of major medical conditions and mortality rates in the THIN cohort is generalizable to the U.K. population [9]. The database consists of coded information on patient demographics, symptoms and diagnoses, drug prescriptions, consultations, diagnostic tests and their results. The diagnoses in the THIN database are recorded using a hierarchical system called Read codes used to describe a health-related concept in GPs’ records [10]. Collection of data for THIN was approved by the South-East Multicentre Research Ethics Committee in 2003. Approval for the use of THIN data in this analysis was obtained from the Scientific Review Committee in April 2018 (SRC reference18THIN001).

### 2.2. Study Population 

All data included in this study were from practices that met the acceptable mortality reporting (AMR) and acceptable computer usage (ACU) standards measures of quality assurance for THIN data [11,12]. Adult patients who had a value for circulating concentrations of serum folate from 1 Jan 1990 to 15 May 2017 were included. In order to avoid competition from comorbidities in the older patient population, those 70 years and older were excluded from the analysis. To limit the effects of folate being measured because patients were being investigated for suspected peripheral neuropathy, the main analyses excluded all person-years, diagnoses of peripheral neuropathy and deaths in the first year of follow-up. Some patients had serum folate concentrations measured on multiple occasions but only the first measurement was used for this analysis. A total of 1,052,622 had a measurement of serum folate. Patients were excluded if they had prevalent peripheral neuropathy (*n* = 138,903), had a value of serum concentrations of folate that was above 68.2 nmol/L or not recorded (*n* = 76,854), or a missing BMI (*n* = 2387) or were older than 70 years (*n* = 240,140) leaving 594,338 patients available for this analysis (Figure 1).

### 2.3. Exposures

Folate deficiency was defined as serum folate <6.8 nmol/L, folate insufficiency was defined as serum folate 6.8 to <13.6 and folate sufficiency was defined as serum folate ≥13.6 nmol/L based on World Health Organization (WHO) guidance [13]. 

### 2.4. Outcome

Peripheral neuropathy was the primary outcome and was defined using Read codes that included peripheral neuropathy or polyneuropathy but excluded cause-specific peripheral neuropathy, such as diabetes and alcoholism. The secondary outcome for this analysis was death from any cause. 

### 2.5. Statistical Analysis

Person-years of follow-up were calculated from one year after the date of measurement of serum folate concentrations up to whichever came first: diagnosis of peripheral neuropathy, exit from the THIN database (transferred practice or died), the last date practice data was collected or December 31 2017. Cox proportional hazard regression was used to calculate the hazard ratios (HRs) and 95% confidence intervals (CI) of peripheral neuropathy and mortality by categories of serum folate concentrations. As there was a statistically significant folate by age interaction, analyses were conducted separately for the following age categories: <40, 40–55, 56–70 years. All analyses were adjusted for sex; age; fifths of socioeconomic group (based on the Townsend deprivation score), smoking (never, past, current, missing); BMI (<25, 25 to <30, ≥30 kg/m^2^); vitamin B_12_ deficiency [13]; use of methotrexate; alcoholism; and chronic liver disease, diabetes, thyroid dysfunction, renal failure, human immunodeficiency virus (HIV) and rheumatoid arthritis. Patients with missing data on covariates were assigned to a separate category for that variable and included in the regression analyses. The p-value for trend of an association between serum concentrations of folate and risk of peripheral neuropathy and mortality was taken from the Cox proportional hazard coefficients. It is possible that undiagnosed disease may cause people to become unwell, which can result in some individuals changing their diet, and may lead to reverse causation bias whereby low serum folate are a consequence, rather than a cause of the disease. To minimise this type of bias, analyses for peripheral neuropathy and mortality were repeated by censoring an additional two years of follow-up.

All statistical analyses were performed using R software [14]. Two-sided *p*-values <0.05 were considered statistically significant. 

## 3. Results

There were 390,675 women and 203,663 men with a measurement of serum folate concentration included in this analysis. Table 1 shows the characteristics of the patients and details of their follow-up according to the categories of baseline serum concentrations of folate. Among all patients, the mean serum folate concentration was 18.7 (SD = 10.7) nmol/L. Across increasing categories of serum folate, the percentage who were women increased as did the average age. Those with a higher serum folate were more likely to be from the least deprived areas, have a BMI in the normal range and were less likely to be current smokers. Across the increasing categories of folate, serum concentrations of vitamin B_12_ also increased.

After a mean follow-up of 3.7 (SD = 3.1) years, there were 1949 patients diagnosed with peripheral neuropathy (947 women and 1002 men) and 20,679 patients died (9695 women and 10,984 men). The incidence of peripheral neuropathy was higher in men (0.49%) than in women (0.24%).

Figure 2 shows the unadjusted cumulative hazard of peripheral neuropathy for the three age groups according to serum concentrations of folate. 

After adjusting for age, sex and other confounding variables, there was an inverse relation between serum folate concentrations and peripheral neuropathy among those <40 years of age. The adjusted hazard ratios with baseline folate >13.6 nmol/L set as a reference within each age group are shown in Table 2. There was a dose-response association between serum folate and the risk of peripheral neuropathy in participants aged 18–40 years old; those with low-medium folate (6.8–13.6 nmol/L) and low folate (<6.8 nmol/L) had a HR 1.83 (95% CI = 1.16–2.91) and 1.48 (95% CI = 1.04–2.08), respectively, greater hazard of peripheral neuropathy compared to patients with normal serum folate. The strength of the association between serum folate and hazards of peripheral neuropathy was not significant among participants aged 41–55 years of age (*p*-trend = 0.35) and aged 56–70 years (*p-*trend = 0.31). 

The inverse association between serum folate concentrations and the risk of mortality was evident across the three age groups where HR in the lowest folate category compared with the highest folate category was 1.55 (95% CI = 1.26–1.90) in those aged 18–40 years, 1.66 (95% CI = 1.51–1.83) in those aged 41–55 years and 1.48 (95% CI = 1.40–1.57) in those aged 56–70 years.

All analyses were repeated after censoring an additional two years from the index data and the results are shown in Table 3. The inverse association between lowest folate and peripheral neuropathy at ages 18–40 years was attenuated (HR = 1.49; 95% CI = 0.90–2.46) and folate 6.8–13.6 nmol/L (HR = 1.39; 95% CI = 0.97–1.98, *p-*trend = 0.05). Censoring of an additional two years of follow-up made little difference to the association of serum folate and hazards of mortality.

The correlation between vitamin B_12_ and folate concentrations as measured using Pearson’s correlation was *r* = 0.24. There was no overall association between serum vitamin B_12_ concentrations and the risk of peripheral neuropathy. 

## 4. Discussion

For the patients 40 years and younger, circulating folate concentrations were inversely associated with the risk of developing peripheral neuropathy in a dose-dependent manner. There was also a significant inverse association between serum folate and mortality across all age groups. 

Only a few observational studies have examined the relationship between serum folate and the risk of peripheral neuropathy; results from one small cross-sectional study showed that in patients with Parkinson’s disease there was an inverse association between serum folate and the risk of peripheral neuropathy that was independent of age [15]. The results from this study are difficult to interpret since serum folate concentrations were measured at around the time of disease diagnosis, and patients may have changed their diet because of early symptoms, before the formal diagnosis of peripheral neuropathy. There have been several small case-series reports of patients with neuropathy and folate deficiency [7,16]. In the first such study, the symptoms of 8 out of 10 patients (80%) with neuropathy and isolated folate deficiency were either reversed, or they responded to folate treatment [7]. Yukawa et al. [17] showed that the neurological symptoms improved after folic acid supplementation in 24 out of 36 (67%) patients with concomitant folate deficiency and neuropathy. Whereas, Koike et al. [16] reported that only 5 out of 18 (28%) folate deficient neuropathy patients had some improvements in their symptoms after supplementing with folic acid. While the results from these studies may appear compelling, the lack of a comparison group makes it difficult to establish whether the folate treatment caused the improvement in symptoms or symptoms may have naturally resolved over time. Results from a systematic review of randomised or quasi-randomised trials showed that B vitamins were not effective at treating peripheral neuropathy in patients with diabetes or alcoholism but none of the trials’ supplements included folic acid [18]. In the absence of large randomised controlled trial evidence, the association between serum folate and peripheral neuropathy should be examined in other large longitudinal studies. It is also possible that the imminent introduction of folic acid fortification of flour in the U.K. [8] will act as a natural experiment to evaluate not only the effect on the incidence of neural tube defects, but also the effect on the incidence of peripheral neuropathy. It will be of particular importance to examine whether any change in serum folate and incidence of peripheral neuropathy differs according to age as the results from this study suggest. The findings reported herein showed that the association between serum folate and peripheral neuropathy was weaker after excluding the earlier years of follow-up and the possibility that this may be due to reverse causation should be examined in other longitudinal studies with a longer follow-up time.

Peripheral neuropathy is much more common among patients with vitamin B_12_ deficiency because of the role that vitamin B_12_ has as a co-enzyme in the conversion of homocysteine to methionine, a precursor to S-adenosyl methionine (SAM), which is important for myelination of nerve cells [19]. As nerve cells may have some ability to accumulate folate, the risk of neuropathy in those with folate deficiency is meant to be much rarer than that of vitamin B_12_ deficiency [20]. Nevertheless, because of the essential role of folate in the methylation of homocysteine to methionine [19], together with evidence from this and other studies, deficiency in folate may pose a much greater risk for the development of peripheral neuropathy, particularly among younger people, than previously thought. These results suggest that clinicians should continue to measure serum folate in patients suspected of having peripheral neuropathy. 

The inverse association between serum folate and risk of all-cause mortality has also been reported in one other large cohort study conducted in the USA where flour has been mandatorily fortified with folic acid since 2002 [21]. Despite the strong association between serum folate and mortality in our study, an individual patient meta-analysis of randomised controlled trials showed that the risk of dying among participants allocated to folic acid supplementation (either alone or in combination with vitamin B_6_ and/or vitamin B_12_) was not significantly different from the control group [22]. This is further supported by the results from a large trial in China of folic acid supplementation that was not included in the meta-analysis [23]. While the inverse association between serum folate and mortality in this analysis was quite strong, it is possible that it is the result of residual confounding or confounding by unmeasured variables [24]. 

This study has some limitations. From the patients’ records, we were not able to determine the method used to measure serum folate concentrations, which is important as methods, such as folate binding assays, produce systematically lower folate results than the microbiological assay or the gold standard liquid chromatography coupled to tandem mass spectrometry [25]. However, the mean serum folate concentration from this patient population of 18.7 nmol/L is only slightly lower than that of men and women age 19 to 64 years from the 2008/09–2011/12 National Diet and Nutrition Survey of 19.8 nmol/L [26]. This study may be vulnerable to bias since tests for folate may be requested in response to particular symptoms, such as distal paraesthesiae or pain (possibly suggestive of neuropathy), despite the absence of an established neuropathy diagnosis at the time of testing. To overcome this source of bias, the entire first year of follow-up was censored in the main analysis. The neuropathy subtype was also not considered here, and although sensory axonal neuropathies are the most commonly observed, there was no detailed clinical data available. The current analysis used only the first measurement of serum folate and this might not reflect long-term or usual values. Several studies of serum folate in blood samples taken one to four years apart have reported intra-class correlations between 0.5 and 0.6 [27,28,29]. This could mean that the associations of serum folate and peripheral neuropathy risk could be stronger than those reported here. Finally, the population studied has a larger proportion of women than the general U.K. population, potentially limiting the generalisability of these findings.

In conclusion, our results suggest that folate deficiency is a risk factor for neuropathy in patients younger than 40 years. Importantly, the risk of peripheral neuropathy increased as serum folate decreased, and even insufficient serum folate of 6.8 to 13.5 nmol/L appeared to be important. These associations warrant further investigation in other large electronic health datasets. Moreover, future surveillance of peripheral neuropathy in the U.K. after the introduction of folic acid fortification of flour may help to understand whether improving serum folate concentrations in the population also leads to a change in the incidence of peripheral neuropathy across the different age groups.

## Figures and Tables

**Figure 1 nutrients-11-02443-f001:**
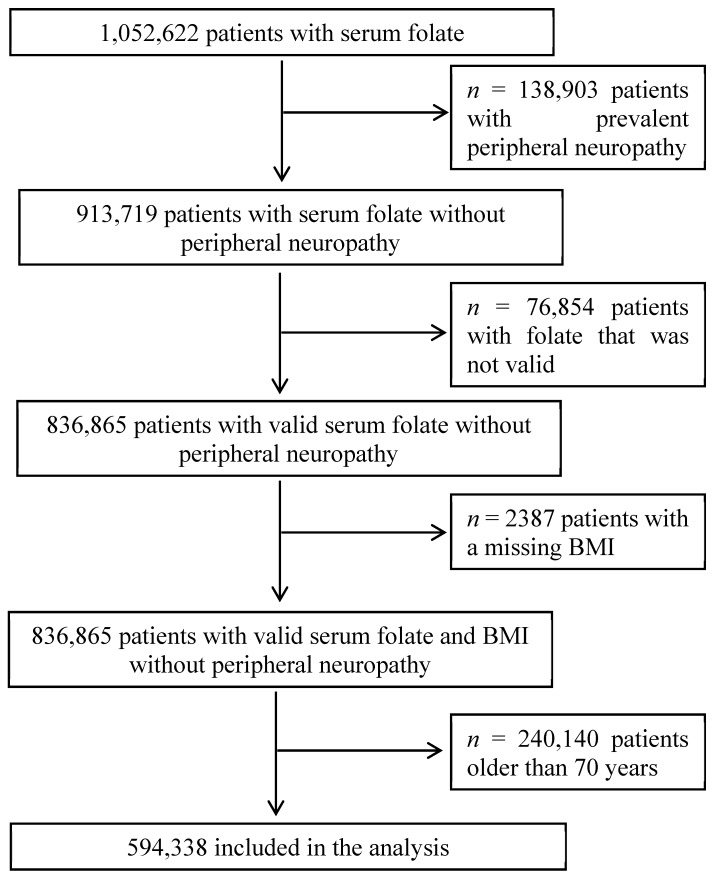
Flow diagram of patients included for an association between serum concentrations of folate and the risk of peripheral neuropathy and mortality.

**Figure 2 nutrients-11-02443-f002:**
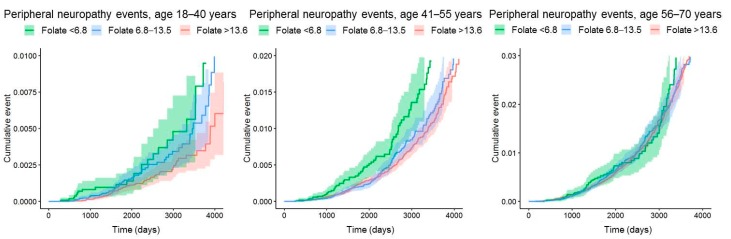
Cumulative event curves for peripheral neuropathy according to serum folate concentrations and age, using 95% Kaplan–Meier confidence interval (CI). Time in days (d).

**Table 1 nutrients-11-02443-t001:** Characteristics of THIN patients according to categories of serum concentrations of folate.

		Serum Concentrations of Folate (nmol/L)		
	<6.8	6.8–13.5	≥ 13.6	*p*-Value
*n*	38,574	187,550	368,214	
Age (years), mean (SD)	45.1 (14.2)	46.1 (14.2)	48.9 (13.5)	<0.001
BMI (kg/m^2^), mean (SD)	28.0 (7.4)	27.8 (6.7)	26.8 (5.7)	<0.001
Men, *n* (%)	13,451 (34.9%)	66,400 (35.4%)	123,812 (33.6%)	<0.001
Age group, *n* (%)				
18-40 years	14,696 (38.1%)	66,681 (35.6%)	104,970 (28.5%)	<0.001
41-55 years	13,378 (34.7%)	65,946 (35.2%)	129,547 (35.2%)	
56-70 years	10,500 (27.2%)	54,923 (29.3%)	133,697 (36.3%)	
Townsend deprivation index				
1 (least deprived)	4630 (12.8%)	30,899 (17.5%)	81,690 (23.5%)	<0.001
2	5073 (14.0%)	29,601 (16.8%)	69,308 (19.9%)	
3	6747 (18.6%)	34,082 (19.3%)	66,832 (19.2%)	
4	7897 (21.8%)	34,503 (19.5%)	55,031 (15.8%)	
5 (most deprived)	8104 (22.4%)	28,824 (16.3%)	36,836 (10.6%)	
Not known	3535 (9.8%)	18,269 (10.3%)	37,286 (10.7%)	
Ethnicity, *n* (%)				
Asian or Asian British	549 (1.4%)	4872 (2.6%)	10,946 (3.0%)	<0.001
Black or black British	329 (0.9%)	2674 (1.4%)	6122 (1.7%)	
White	13,521 (35.1%)	69,216 (36.9%)	133,451 (36.2%)	
Mixed	92 (0.2%)	677 (0.4%)	1381 (0.4%)	
Other	127 (0.3%)	1030 (0.5%)	3487 (0.9%)	
Not known	23,956 (62.1%)	109,081 (58.2%)	212,827 (57.8%)	
Smoking				
Non-smokers	13,043 (33.8%)	89,818 (47.9%)	220,822 (60.0%)	<0.001
Former smokers	7,081 (18.4%)	40,208 (21.4%)	85,197 (23.1%)	
Current smokers	18,101 (46.9%)	56,062 (29.9%)	59,665 (16.2%)	
Not known	349 (0.9%)	1462 (0.8%)	2530 (0.7%)	
Follow-up				
Person-years of follow-up	684,809	576,934	1,353,773	
New cases of peripheral neuropathy	204	598	1147	
Deaths	10,675	2842	7162	

Abbreviations: SD—standard deviation; BMI—body mass index; THIN—The Health Improvement Network.

**Table 2 nutrients-11-02443-t002:** Hazard ratios (95% CI (confidence intervals)) of peripheral neuropathy and mortality according to categories of serum folate concentrations and age group.^1^.

	Serum Concentrations of Folate (nmol/L)			
	<6.8	6.8–13.5	≥ 13.6	*p* for Trend^2^
Peripheral neuropathy				
18–40 years	1.83 (1.16–2.91)	1.48 (1.04–2.08)	1.00 (ref)	0.0043
41–55 years	1.11 (0.85–1.45)	1.06 (0.89–1.27)	1.00 (ref)	0.35
56–70 years	0.91 (0.71–1.17)	0.94 (0.82–1.07)	1.00 (ref)	0.31
Mortality				
18–40 years	1.55 (1.26–1.90)	1.07 (0.91–1.24)	1.00 (ref)	0.0002
41–55 years	1.66 (1.51–1.83)	1.23 (1.15–1.32)	1.00 (ref)	<0.0001
56–70 years	1.48 (1.40–1.57)	1.30 (1.25–1.34)	1.00 (ref)	<0.0001

^1^ Adjusted for sex, age, socioeconomic group, smoking, BMI, vitamin B_12_ deficiency, use of methotrexate, alcoholism, chronic liver disease, diabetes, thyroid dysfunction, renal failure, human immunodeficiency virus (HIV) and rheumatoid arthritis. ^2^ Test for linear trend.

**Table 3 nutrients-11-02443-t003:** Hazard ratios (95% CI) of peripheral neuropathy and mortality according to the categories of serum folate concentrations and age group after censoring an additional two years of follow-up.^1,2^.

	Serum Concentrations of Folate (nmol/L)			
	<6.8	6.8–13.5	≥ 13.6	*p* for Trend ^3^
Peripheral neuropathy				
18–40 years	1.49 (0.90–2.47)	1.41 (0.99–2.01)	1.00 (ref)	0.05
41–55 years	1.16 (0.88–1.53)	1.12 (0.93–1.34)	1.00 (ref)	0.187
56–70 years	0.94 (0.73–1.22)	0.97 (0.85–1.12)	1.00 (ref)	0.588
Mortality				
18–40 years	1.53 (1.19–1.97)	1.07 (0.88–1.30)	1.00 (ref)	0.0032
41–55 years	1.65 (1.47–1.86)	1.31 (1.20–1.42)	1.00 (ref)	<0.0001
56–70 years	1.45 (1.35–1.55)	1.30 (1.24–1.36)	1.00 (ref)	<0.0001

^1^ Adjusted for sex, age, socioeconomic group, smoking, BMI, vitamin B_12_ deficiency, use of methotrexate, alcoholism, chronic liver disease, diabetes, thyroid dysfunction, renal failure, human immunodeficiency virus (HIV) and rheumatoid arthritis. ^2^ Includes 1792 cases of peripheral neuropathy and 13,257 deaths. ^3^ Test for linear trend.

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
