# Peer review of "Circulating Folate Concentrations and Risk of Peripheral Neuropathy and Mortality: A Retrospective Cohort Study in the U.K"

_nutrients, 2019, doi:10.3390/nu11102443_

Round 1
Reviewer 1 Report
The topic is of great interest. The influence of B-vitamins on the various neurological disturbance mechanisms is a hot topic that needs, however, more updated data. The large population sample represents a strength of the manuscript which in its simplicity and linearity deals exhaustively with the topic.
Some aspects should be improved which, in my opinion, could make the article more fluid in reading.
- Since the folate deficiency cut-off has been defined, the range 6.8-13.5 could be defined as a range of insufficiency
- Supplementary Figure 1 could be inserted in the text as Figure 1 and, in any case not repeated in the text and also in the supplementary material
- The same as above for the Supplementary table 1. This table should not appear in the main text of the manuscript and the supplementary material. In my opinion, it could be moved to page 6 as a regular table, immediately close to the text that refers to it. More generally, there would be no need to put these two elements in the supplementary material and, if preferred, should not be repeated in the main text of the manuscript.
- About this table, the meaning and experimental utility of the omission of the first two years of follow-up should be better clarified in the text. Why was this analysis performed and what implications do the results show?
- Regarding the exclusion criteria, the BMI range 14-75 seems to me to be unusual. I assume there is an upper limit error. In this case, it is better to correct it both in the text and in the flow diagram. If it is correct, it should be clarified why it is so non-stringent (almost descriptive) in the upper limit. What are the maximum and minimum values of the population sample about BMI?
- In the first paragraph on page 6, there are problems with text formatting (font size and leading)
- The first lines on page 8, talk about the omission in the analysis of the first year of follow-up. I think this was done to exclude those cases with manifested neuropathy in this time frame that underwent evaluation of serum folate for the suspicion of this pathology. In addition to being able to broaden this concept better, it is not clear whether this censoring was made in the final analysis or whether an additional analysis was made without obtaining conflicting results with the final ones. In any case, it should be better explained, considering that an additional analysis with the omission of the first two years of follow-up was proposed and that in this case the statistical significance had been lost for HR of peripheral neuropathy.
- The numbering of references at the end of the manuscript is repeated twice.
Reviewer 2 Report
This is an interesting retrospective study investigating the associations between folate deficiency and peripheral neuropathy and mortality.
Some comments and suggestions:
- Persons >70 years were excluded. Comorbidities may also be a ‘problem’ before age 70. I would argue it would also be interesting to investigate these associations in persons above 70 years, given previous reports about differences or reversed associations in old age.
-Some patients had serum folate concentrations measured on multiple occasions but only the first measurement was used for this analysis. Patients may have been aware of their folate level and may have changed dietary habits. Doctors may also have advised the use of folic acid supplements based on this first measurement. This may may have reduced the associations.
- Were users of folate supplements or supplement prescriptions excluded from the analyses?
- The authors adjusted for many variables. Are these all considered to be true confounding variables, in the sense that they are all associated with folate deficiency and peripheral neuropathy? Unnecessary adjustments may perhaps also lead to underestimation of the results?
- Were the analyses for developing peripheral neuropathy adjusted for competing risks (like mortality)? If not, why not?
Round 2
Reviewer 2 Report
Thank you for your response, explanations and additional analyses. I have no further comments
Author Response
Thank you for your response, explanations and additional analyses. I have no further comments.
We thank the reviewer for taking the time to read over our responses.